# Transcriptional Profiling of Human Endothelial Cells Unveils PIEZO1 and Mechanosensitive Gene Regulation by Prooxidant and Inflammatory Inputs

**DOI:** 10.3390/antiox12101874

**Published:** 2023-10-17

**Authors:** German A. Arenas, Jose G. Valenzuela, Estefanía Peñaloza, Adolfo A. Paz, Rodrigo Iturriaga, Claudia G. Saez, Bernardo J. Krause

**Affiliations:** 1Instituto de Ciencias de la Ingeniería, Universidad de O’Higgins, Rancagua 2841959, Chile; german.arenas@uoh.cl; 2Department of Hematology-Oncology, Pontificia Universidad Católica de Chile, Santiago 8331150, Chilecsaezs@uc.cl (C.G.S.); 3Instituto de Ciencias de la Salud, Universidad de O’Higgins, Rancagua 2841959, Chile; 4Instituto de Ciencias Biomedicas, Facultad de Medicina, Universidad de Chile, Santiago 7500000, Chile; 5Facultad de Ciencias Biológicas, Pontificia Universidad Católica de Chile, Santiago 8331150, Chile; Rodrigo.Iturriaga@uantof.cl; 6Centro de Investigación en Fisiología y Medicina en Altura, Facultad de Ciencias de la Salud, Universidad de Antofagasta, Antofagasta 1271155, Chile

**Keywords:** endothelial, mechanosensing, PIEZO1, NF-kappa B, prooxidants, disturbed blood flow, inflammation, TNF-α

## Abstract

PIEZO1 is a mechanosensitive cation channel implicated in shear stress-mediated endothelial-dependent vasorelaxation. Since altered shear stress patterns induce a pro-inflammatory endothelial environment, we analyzed transcriptional profiles of human endothelial cells to determine the effect of altered shear stress patterns and subsequent prooxidant and inflammatory conditions on PIEZO1 and mechanosensitive-related genes (MRG). In silico analyses were validated in vitro by assessing PIEZO1 transcript levels in both the umbilical artery (HUAEC) and vein (HUVEC) endothelium. Transcriptional profiling showed that PIEZO1 and some MRG associated with the inflammatory response were upregulated in response to high (15 dyn/cm^2^) and extremely high shear stress (30 dyn/cm^2^) in HUVEC. Changes in PIEZO1 and inflammatory MRG were paralleled by p65 but not KLF or YAP1 transcription factors. Similarly, PIEZO1 transcript levels were upregulated by TNF-alpha (TNF-α) in diverse endothelial cell types, and pre-treatment with agents that prevent p65 translocation to the nucleus abolished PIEZO1 induction. ChIP-seq analysis revealed that p65 bonded to the *PIEZO1* promoter region, an effect increased by the stimulation with TNF-α. Altogether this data showed that NF-kappa B activation via p65 signaling regulates PIEZO1 expression, providing a new molecular link for prooxidant and inflammatory responses and mechanosensitive pathways in the endothelium.

## 1. Introduction

Vascular tone regulation is a biomechanical process tightly regulated by local signals, such as endothelial-derived factors, in response to changes in blood flow and shear stress. Shear stress is proposed as the primary physiological inductor of vasodilation, which is mediated by mechanotransduction processes [1]. PIEZO1, a mechanosensitive non-specific cation channel, is implicated in shear stress sensing in endothelial cells [2]. Compelling evidence shows that PIEZO1 channels are critical for vascular structure integrity during development and blood pressure regulation. Indeed, endothelial cell (EC)-specific deletion of PIEZO1 in a murine model induces a lethal phenotype during embryonic development [2] and hypertension in adults, respectively, by impairing nitric oxide (NO)-mediated vasorelaxation [3]. Additionally, growing evidence shows that PIEZO1 is involved in several vascular processes, such as inflammation, angiogenesis, and endothelial cell alignment [4,5,6].

An altered shear stress pattern induces oxidative stress and atheroprone, pro-inflammatory, and pro-thrombotic endothelial cell phenotypes [7]. These disturbed hemodynamic shear forces activate the nuclear factor kappa B (NF-κB) pathway, a family of transcription factors that regulate several critical genes involved in vascular stability [8,9], and mediates the increase in oxidative stress and inflammatory cytokine signaling, such as tumor necrosis factor alpha (TNF-α) [10]. In this regard, inflammatory mediators play a crucial role in the effects resulting from disturbed blood flow patterns. TNF-α initiates pro-inflammatory signaling, mainly by activating the NF-κB complex transcriptional function, leading to endothelial inflammation and arterial wall injury [11,12]. Furthermore, it has been demonstrated that Krüppel-like factor 4 (KLF4), a mechanoresponsive transcription factor, regulates the promoter activity of anti-inflammatory and antioxidant genes in response to physiological shear stress [13]. Similarly, yes-associated protein (YAP) plays a key role in mechanotransduction in response to extracellular biophysical cues (i.e., cell stiffness) participating in critical physiological processes, such as vascular development [14].

Although a growing body of evidence supports that PIEZO1 mediates many physiological processes, little evidence addresses how its expression is affected and regulated by shear stress-responsive transcription factors, and whether changes in PIEZO1 expression correlate with other mechanotransduction-related gene expression (MRG) [15]. To tackle this gap, this work aims to identify potential transcriptional regulators of PIEZO1 and MRG in endothelial cells by combining in silico analysis of transcriptomic datasets and in vitro validation in the human arterial and venous endothelium. Findings on endothelial PIEZO1 regulators will shed light on the pathways and interactions involved in regulating this novel mechanosensitive protein, which has a considerable impact on vascular function in health and disease.

## 2. Materials and Methods

### 2.1. Experimental Design

Transcriptomic datasets were selected as previously described [16,17], applying PRISMA EQUATOR guidelines for the meta-analysis studies (http://www.equator-network.org/reporting-guidelines/prisma, accessed on 30 June 2021). Using “Endothelial cells”, “HUVEC”, “TNF-α”, and “Affymetrix” as keywords, we explored GEO (Gene Expression Omnibus, http://www.ncbi.nlm.nih.gov/geo/, accessed on 30 October 2021; and OmicsDI, http://www.omicsdi.org, accessed on 30 October 2021). The data available up to October 2021 were considered in this analysis. Appendix A summarizes the datasets with accession numbers and downloadable URLs. Each available dataset was identified with its corresponding GEO number, containing detailed information for each experiment. Raw data contained in CEL files were analyzed using the software TAC (Transcriptome Analysis Console) 4.0.2.15 version (Applied Biosystems, Waltham, MA, USA), gathering datasets with the same Platform ID. For differentially expressed gene (DEG) analysis, the following cut-off values were applied: −1.5 < fold change > 1.5; a *p*-value < 0.05 and FDR < 0.05 using the Limma package with the eBayes method for multiple comparisons. 

### 2.2. Functional Enrichment

Biological process, BP (BP), and KEGG pathway enrichment analysis were performed in the GeneOntology (GO) (http://geneontology.org/, accessed on 30 June 2022) [18] and Enrichr (https://maayanlab.cloud/Enrichr/, accessed on 30 June 2022) [19] databases. False Discovery Rate (FDR)-adjusted *p*-values (*p* < 0.05) and their relationship with general vascular function and cell differentiation were used as the main criteria for the selected terms.

### 2.3. Definition of Mechanotransduction-Related Genes (MRG)

To identify and define genes involved in mechanotransduction in endothelial cells that contribute to shear stress signaling and NO-dependent relaxation, a brief literature review was performed. Twenty-seven genes were selected, considering the evidence supporting their contribution to mechanical stimuli and NO transduction. (Appendix A). Enrichment analysis was performed using selected genes to establish which pathways (“term”) were represented. The analysis showed that “Fluid shear stress and atherosclerosis” obtained the highest combined score (~4133). In the same way, other terms related to inflammatory pathways (TNF-α—NF-κB signaling pathway) and PIEZO1 signaling (Akt—signaling pathway) obtained high combined scores and were included in a summary table (Appendix A), supporting the representation of these 27 genes in endothelial flow-related mechanotransduction.

### 2.4. Transcription Factor Enrichment Prediction

Transcriptional regulators of MRG were predicted using the TFEA.ChIP package (https://www.iib.uam.es/TFEA.ChIP/, accessed on 31 March 2023), which analyzes transcription factor enrichment in a gene set using data from ChIP-seq experiments [20], available at ENCODE project, GEO Datasets (GeneHancer ReMap 2018). The pre-defined list of MRG was submitted for analysis, selecting Human as the source organism, and ReMap2020 and GeneHancer Double Elite as the ChIP-seq source.

### 2.5. Clustering, Heatmap, and Chord Plot Visualization

Hierarchical clustering was created using the R-based web tool, Morpheus (Versatile matrix visualization and analysis software) (https://software.broadinstitute.org/morpheus, accessed on 31 October 2022). Filtered datasets were uploaded in Morpheus, and hierarchical clustering was created based on the average expression of genes in each dataset. The heatmap visualization was created in GraphPad Prism 8.0 (GraphPad Software Inc., San Diego, CA, USA), based on the average expression of mechanosensing-related genes considering statistical differences when the False Discovery Rate (FDR) adjusted *p*-value is *p* < 0.05. T-distributed Stochastic Neighbor Embedding (t-SNE) dimensionality reduction was used to determine the similarity among samples in transcriptional terms. The samples were clustered by rows (samples) and columns (transcripts) using Euclidean distance and embedded in the first two axes of the t-SNE space.

### 2.6. Transcription Factor Binding Sites Prediction

Promoter-associated transcription factor binding site prediction was performed using MatInspector, an online software that is based on sequences using a position weight matrix (complete nucleotide distribution in every single position) [21]. NF-κB binding site prediction to PIEZO1 promoter regions was performed in the Homo sapiens genome, considering 0.9–1.0 as a high score matrix (% coincidence). The data were filtered with a score of coincidence of ≥0.8 between the promoter sequence and TF sequence for highly probable predictions (Appendix A).

### 2.7. ChIP-seq Profiling and Visualization

The ChIP-seq dataset (GSE53998) was retrieved from the Gene Expression Omnibus (www.ncbi.nlm.nih.gov/geo/, accessed on 30 July 2022). Handling and visualization of the ChIP-seq dataset were done using EaSeq software v1.12 (http://easeq.net, accessed on 30 July 2022) [22]. The PIEZO1 promoter region was identified according to hg18 coordinates (chr16:87378667—87382433), and the analyzed data considered polymerase II (POLII) and v-rel reticuloendotheliosis viral oncogene homolog A (RELA) chromatin immunoprecipitation experiments. The promoter region considered starts of −2000 bp from the transcription start site (TSS). The qualitative graph shows the PIEZO1 gene promoter area and TSS in the opposite direction of transcription.

### 2.8. Human Umbilical Endothelial Cell Cultures

Human umbilical artery (HUAEC) and vein (HUVEC) endothelial cells were purchased from Sigma (Sigma Aldrich, Merck KGaA, Darmstadt, Germany). The cells were cultured in a Growth Medium MV2 Kit (Sigma Aldrich, Merck KGaA, Darmstadt, Germany) with Microvascular Growth Supplement (MVGS) S00525 (Invitrogen Waltham, MA, USA) in standard culture conditions (Air + 5% CO_2_, 37 °C) [23]. Upon confluence, the cells were plated up to the fifth or sixth passage, and then serum-starved (2% MVGS) and exposed to the different experimental conditions. Human TNF-α (Thermo Fisher, Waltham, MA, USA) was diluted in a starving cell medium (2% MVGS) at 50 ng/mL. The cells (HUAEC–HUVEC) were washed three times with sterile PBS 1× buffer (137 mM NaCl, 2.7 mM KCl, 10 mM Na2HPO4, and 1.8 mM KH2PO4). After washing, starved serum (2 mL) and TNF-α (50 ng/mL) were added for 24 h. 

### 2.9. Isolation of RNA

Total RNA containing the mRNA fraction was isolated using a modified Chomczynski *p* protocol by Trizol (Thermo Fisher, Waltham, MA, USA), according to the manufacturer’s instructions [24]. RNA concentrations were calculated based on the absorbance at 260 nm. The RNA samples were converted to cDNA, and the rest were stored at −80 °C until use.

### 2.10. Quantification of mRNA Levels

The cDNA synthesis for qPCR was transcribed through a OneScript cDNA Synthesis Kit (ABM, Richmond, ON, Canada). The levels of mRNA were quantified through quantitative real-time PCR (qRT-PCR) in StepOne Plus (Applied Biosystems), using KicqStart Kit (Sigma Aldrich, Merck KGaA, Darmstadt, Germany) with specific primers (sequences in Appendix A). All the procedures were conducted according to the manufacturer’s protocol. The relative expressions were calculated using the 2−ΔΔCt method [25]. The control transcripts used for 2−ΔΔCt were ATP5F1 and RPLP2 as a housekeeper for the mRNA [16,23].

### 2.11. Quantification and Statistical Analysis

All the values were expressed as a mean ± statistical standard error of the mean (S.E.M), where n indicates the number of individual cultures. Comparisons between the two groups were performed by a non-parametric Mann–Whitney U test. Statistical analyses of treatments and comparisons were performed by one-way ANOVA, as appropriate. If the ANOVA demonstrated a significant interaction between variables, post hoc analyses were performed by the false discovery rate (FDR). In all the analyses, *p* < 0.05 was considered to represent statistical significance. All the statistical analyses were conducted using GraphPad Prism 8.0 (GraphPad Software Inc., CA).

## 3. Results

### 3.1. Effect of High Shear Stress on PIEZO1 and Mechanotransduction-Related Gene (MRG) Transcript Levels in HUVECs

To explore the effect of high shear stress on PIEZO1 transcript levels, we analyzed the GSE46248 dataset. Hierarchical clustering of HUVECs samples exposed to static (“Ctrl”), 15 dyn/cm^2^ (“15 dyn”, high shear stress), and 30 dyn/cm^2^ (“30 dyn”, extremely high shear stress) showed clear segregation according to the gene expression profiles in each condition (Figure 1A), with a high correlation in the gene expression patterns of cells exposed to the high (15 dyn/cm^2^) and extremely high (30 dyn/cm^2^) shear stress levels (Figure 1B). The transcriptional profiles associated with the high and extremely high shear stress levels were enriched in genes regulated by atheroprone factors according to the TFEA ChIP analysis, with several hints of RELA (n = 82), and lower representation of KLF4 (n = 6) and YAP1 (n = 7) (Figure 1C). Interestingly, high and extremely high shear stress levels upregulated PIEZO1 transcripts to comparable levels (Figure 1D), an effect also observed in 16 out of the 27 MRG transcripts (Figure 1E).

### 3.2. Effects of Shear Stress-Responsive Gene Silencing on PIEZO1 and MRG Transcript Levels in HUVEC

To determine the potential regulation of KLF4 and YAP1 pathways on PIEZO1 and MRG transcripts, two independent available datasets (GEO numbers: GSE61989; GSE32693) from knocking-down experiments were analyzed. The silencing of KLF4 induced changes in 8 out of the 27 MRG (Figure 2A), with no changes in PIEZO1 transcript levels (Figure 2B). Similarly, a small proportion of MRG transcripts (3 out of 27) were differentially expressed in HUVEC treated with siRNA YAP1 (Figure 2C), with null effects on PIEZO1 transcript levels (Figure 2D), suggesting that neither KLF4 nor YAP1 is directly involved in PIEZO1 regulation.

### 3.3. Effect of TNF-α on PIEZO1 and MRG Transcript Levels in HUVEC

Based on the RELA enrichment, along with the increased expression in PIEZO1 found in HUVEC exposed to high shear stress levels, we aimed to determine the effect of a pro-inflammatory stimulus on MRG transcript levels by combining diverse datasets (GEO: GSE22175, GSE54000, GSE34059) in which treatments with TNF-α in different concentrations and time were applied. The fold change heatmap showed that up to 15 out of the 27 MRG were differentially expressed in HUVEC treated with TNF-α (Figure 3A), especially those related to pro-inflammatory signaling pathways. Additionally, an enrichment in oxidative stress-related pathways was found (Appendix A). TNF-α treatment consistently—at different concentrations and times of treatment—upregulated PIEZO1 transcript levels (Figure 3B). Transcriptomics analysis of time-lapse responses of different MRG and PIEZO1 expressions to TNF-α (10 ng/mL) treatment (GEO: GSE9055) showed that PIEZO1 and RELA transcripts were regulated in parallel, with no clear associations between PIEZO1 and KLF4, nor YAP1 (Figure 3C). Correlation analysis showed high similarity in PIEZO1 and RELA transcript level regulation in response to TNF-α (Figure 3D), supporting the interaction and regulation of both after pro-inflammatory stimulation.

### 3.4. Effect of TNF-α on PIEZO1 and MRG Transcript Levels in Diverse EC

Considering the venous phenotype of HUVEC and the differential regulation of key vascular genes that may take place across the vasculature, the transcriptional effect of TNF-α on PIEZO1 and MRG was assessed using available datasets of endothelial cells from diverse vascular origins. TNF-α treatment (red, orange, and pink dots) conferred a distinctive gene expression profile to endothelial cells (Figure 4A,B), with a remarkable enrichment in NF-κB signaling and fluid shear stress-related pathways (Figure 4C). Changes in PIEZO1 and MRG transcript levels induced by TNF-α treatment were confirmed in cells from different vascular beds (GEO: GSE156116, GSE2638, GSE138991) (Table 1) suggesting a potential role of pro-inflammatory stimulus in PIEZO1 transcript level regulation in the endothelium.

To experimentally validate the effect of TNF-α on PIEZO1 expression, and the occurrence of this response in both arterial and venous endothelium, HUAEC and HUVEC were treated with TNF-α, 50 ng/mL, for 24 h. PIEZO1 transcript levels were increased in both arterial and venous endothelial cells treated with TNF-α (Figure 4D,E), supporting the results obtained by in silico analysis of available datasets.

### 3.5. Interaction of PIEZO1/RELA and the Effects of NF-κB Inhibitors on TNF-α-Induced PIEZO1 Expression

To further study the role of the TNF-α/ NF-κB pathway on PIEZO1 transcriptional regulation, the occurrence of binding sites for NF-κB and/or RELA/p65 subunit within the PIEZO1 promoter region was predicted using MatInspector, resulting in seven highly specific sequences for NF-κB of which two were specific sequences for RELA/p65 (Figure 5A, Appendix A). The latter was confirmed by ChIP-seq data derived from HUVEC (GEO: GSE34500), in which TNF-α induced a robust increase in the interaction of RELA (Figure 5B) but not in the interaction of Pol II (Figure 5C). The requirement for p65 in the TNF-α-dependent induction of PIEZO1 was further addressed in HUVEC pretreated with NOD and CORM-3 (GEO: GSE34059; GSE22175). p65 subunit inhibitors prevented the upregulation of PIEZO1 transcripts (Figure 5D,E). In contrast, no changes in PIEZO1 transcripts were observed when the p65 chaperone, BRD4, was inhibited with JQ1 (GSE54000) (Figure 5F), suggesting a potentially crucial role of p65 in the TNF-α-dependent PIEZO1 upregulation.

## 4. Discussion

We studied regulatory factors coordinating the expression of mechanosensing genes in endothelial cells with a focus on Piezo1, a novel and key mechanosensitive protein within the vasculature and diverse cell types. The results showed that high shear stress and TNF-α treatment, but not KLF4 nor YAP1 silencing, regulated PIEZO1 transcript levels. Pro-inflammatory-mediated induction of PIEZO1 occurred in endothelial cells from diverse vascular beds, and this upregulation was related to the increased interaction of p65 with the PIEZO1 promoter region, which was further prevented by p65 inhibitors. Proof of concept experiments confirmed in vitro PIEZO1 transcript upregulation in both arterial and venous human endothelial cells. Consequently, this study provides novel mechanisms related to PIEZO1 regulation, which contributes to the understanding of shear stress effects on physiological and adverse conditions of the endothelium. 

Several studies have addressed the regulation of mechanosensory components in the endothelium in response to shear stress [1,26]. For instance, KLF2 and KLF4, members of the Krüppel-like factor family, regulate various proteins that respond to mechanical stress [13]. Blood flow also regulates the transcriptional activity and subcellular localization of YAP [13], with effects on blood vessel stability. Additionally, extensive evidence shows that endothelial cell ion channels also play an essential role in response to shear stress profiles, by modifying their expression, permeability, and ion flux [27]. In this regard, we aimed to determine the coordinated regulation of MRG, including Piezo1 as a novel and crucial sensor of flow in the endothelium, in cells exposed to shear stress. Our analysis showed that high shear stress regulates transcript levels from several MRG, including PIEZO1 in HUVEC, with considerable similarities in the profiles under 15 dyn/cm^2^ and 30 dyn/cm^2^ stimulation. Additionally, the transcriptional profiles resulting from the exposure to high shear stress levels were mainly enriched in genes regulated by p65, and a lower representation of KLF- and YAP1-related genes. In this regard, KLF4 and YAP1 silencing showed no effect on PIEZO1 transcript levels, suggesting that these transcription factors do not contribute to PIEZO1 regulation, at least in basal and high shear stress conditions. 

Compelling data support that altered blood flow patterns induce pro-inflammatory states in endothelial cells of various origins [7], affecting mechanosensing proteins [28]. In addition, the response to altered flows increases the expression of adhesion molecules (i.e., VCAM-1/ICAM-1) [29], the production of reactive oxygen species (ROS) [30], and the activation of the pro-inflammatory transcription factor, NF-κB, which commands the inflammatory response in endothelial cells [31]. In this context, transcriptional analysis of HUVEC exposed to high shear stress levels supported a prominent regulation of pro-inflammatory pathways with enrichment in genes regulated by the NF-κB subunit, p65. To further address the contribution of this pathway, we assessed the effect of the pro-inflammatory stimuli (TNF-α) on MRG and PIEZO1 transcript levels. The latter effect was further supported by the in vitro validation, using endothelial cells from arterial and venous origins, in which the PIEZO1 transcript level was induced by TNF-α. Furthermore, in silico analysis showed that TNF-α stimulation resulted in transcriptional changes in pathways related to inflammation, mechanoresponsive transcription factors, and vasodilator agents, including increased PIEZO1 transcript levels. Several studies have demonstrated that TNF-α stimulation contributes to endothelial–mesenchymal transition [32] and a pro-adhesive phenotype [7,15]. Notably, the transcriptional profile associated with TNF-α stimulation was comparable among different endothelial cell types, despite their considerable phenotypic differences [16,17,33], suggesting that this mechanism may affect, in a concerted manner, the endothelial function. 

A recent study shows that RELA (i.e., p65) overexpression in human pulmonary artery endothelial cells induces upregulation of PIEZO1; meanwhile, p65 silencing downregulates PIEZO1 [34]. In this study, we added detailed evidence concerning the regulatory mechanisms of PIEZO1 by the pro-inflammatory and prooxidant transcription factor NF-κB. Our results showed that PIEZO1 induction by TNF-α occurred in association with positive regulation of RELA, with a high correlation between PIEZO1 and RELA transcript levels in a time-lapse analysis. TNF-α stimulation induces NF-κB nuclear translocation and binds to pro-inflammatory responsive genes promoter region zone [35]. Moreover, shear stress disturbances in vascular regions increase NF-κB expression [36], and TNF-α mimics this effect due to activating pro-inflammatory genes and downregulating anti-inflammatory genes [30]. Similarly, our data showed the presence of highly specific binding sites for NF-κB in the human PIEZO1 promoter, as well as increased binding of p65 in endothelial cells stimulated with TNF-α. The regulation of PIEZO1 by p65 was further confirmed by the effects of NF-κB inhibitors, which act at different levels of its signaling pathway. For instance, NOD and CORM-3 inhibit p65 phosphorylation at Ser246 [37] and decrease its levels [38], respectively. Meanwhile, JQ1 inhibits BRD4, a coactivator that enhances NF-κB complex transcriptional activity [39]. Consequently, we showed that the induction of PIEZO1 by TNF-α was exclusively blocked by agents preventing the activation and translocation of p65. Altogether, these data confirm an interaction between RELA–PIEZO1 expression in a pro-inflammatory environment, unveiling a new potential role of PIEZO1 in endothelial inflammation [15].

Some datasets included in this analysis have a small number of samples, which may limit the potential interpretations of the data. Despite this, the integration of a considerable number of independent datasets with different conditions, and the in vitro validation in human endothelial cells from both arterial and venous origins, allow to support the upregulation of PIEZO1 in response to high shear stress and pro-inflammatory conditions. Moreover, this study only considers PIEZO1 transcript level regulation by high shear stress and pro-inflammatory and pro-oxidant environments. Future directions will seek to determine the functional changes in the endothelial PIEZO1 protein under atheroprone conditions and to understand the consequence of these changes in gene expression in an in vivo model that will make these findings clinically translatable.

## 5. Conclusions

In summary (Figure 6), the present data shows that high shear stress increased PIEZO1 and regulated several MRG transcript levels in endothelial cells involving a prooxidant and pro-inflammatory profile. In this context, an inflammatory stimulus increased endothelial PIEZO1 expression, and this upregulation was mediated by RELA/p65. In this context, reactive oxygen species, as key factors signaling the presence of a pro-oxidant environment, may activate the NF-κB pathway, which in turn will result in an up-regulation of PIEZO1 and MRG. Altogether, these results provide novel evidence linking the regulation of PIEZO1 and MRG by altered blood flow patterns, which may contribute to the endothelial dysfunction observed in pro-inflammatory conditions.

## Figures and Tables

**Figure 1 antioxidants-12-01874-f001:**
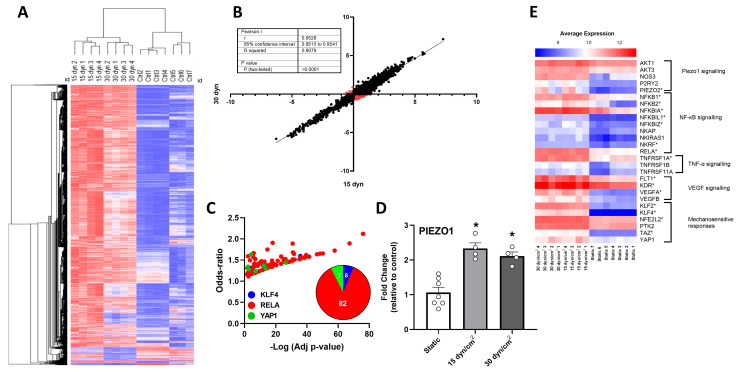
High shear stress regulates PIEZO1 and mechanotransduction-related gene (MRG) levels in HUVEC. (**A**) Hierarchical clustering of HUVEC samples exposed to static, 15 dyn/cm^2^, and 30 dyn/cm^2^ in vitro shear stress. (**B**) Correlation graph for 15 dyn/cm^2^ and 30 dyn/cm^2^ exposed HUVEC. Black dots represent similar gene regulation patterns between groups. Red dots represent different gene regulation between groups. (**C**) Transcription factor enrichment in DEG in HUVEC exposed to 15 dyn/cm^2^ and 30 dyn/cm^2^. (**D**) Fold change in PIEZO1 transcript levels in HUVEC exposed to static, 15 dyn/cm^2^, and 30 dyn/cm^2^ in vitro shear stress (**E**) Average expression heat map for MRG in HUVEC exposed to different shear stress levels. Values are expressed as mean ± S.E.M., * *p* < 0.05, one-way ANOVA, FDR post hoc test.

**Figure 2 antioxidants-12-01874-f002:**
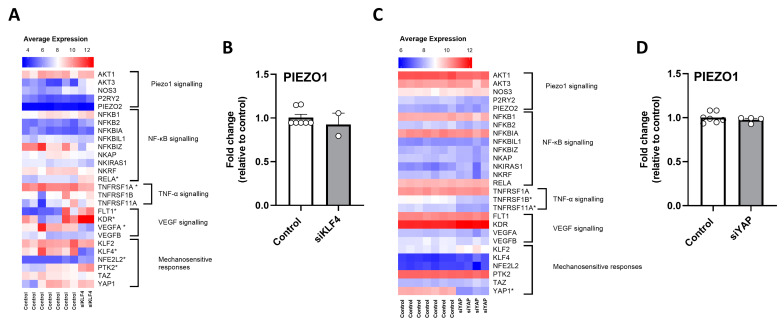
Effect of silencing shear stress-responsive genes on PIEZO1 and MRG transcript levels in HUVECs. (**A**) Average expression heatmap for mechanotransduction-related genes in HUVEC treated with siKLF4. (**B**) PIEZO1 transcript level fold change in HUVEC treated with siKLF4. (**C**) Average expression heatmap for mechanotransduction-related genes in HUVEC treated with siYAP. (**D**) PIEZO1 transcript level fold change in HUVEC treated with siYAP. Values are expressed as mean ± S.E.M., * *p* < 0.05, Mann–Whitney U test.

**Figure 3 antioxidants-12-01874-f003:**
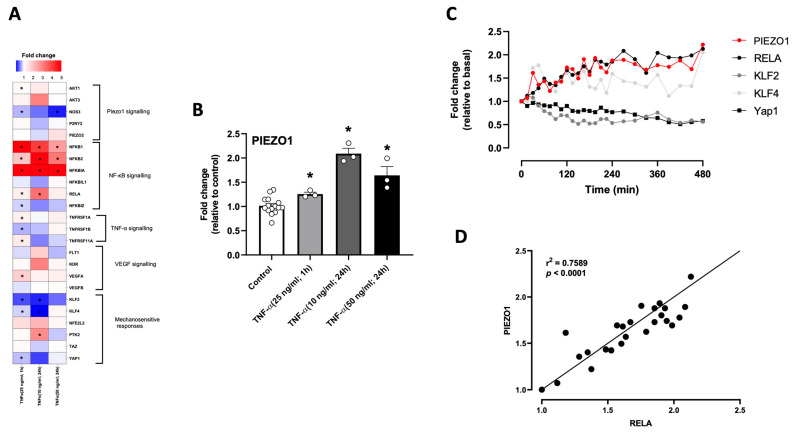
TNF-α increases PIEZO1 and mechanosensing-related transcript levels in diverse EC. (**A**) Fold change heatmap for mechanotransduction-related genes in HUVECs treated with TNF-α, 25 ng/mL (1 h), and 10 and 50 ng/mL (24 h). (**B**) PIEZO1 transcript level fold changes in response to TNF-α treatments at different times and concentrations. (**C**) Time-lapse curves representing selected transcript level changes in response to 480 min of TNF-α (10 ng/mL) treatment. (**D**) Dots graph representing the correlation between PIEZO1 and RELA transcript levels during time-lapse shown in (**C**). Values are represented as fold changes related to the basal from one sample from each data point. Values are expressed as mean ± S.E.M., * *p* < 0.05, one-way ANOVA, FDR post hoc test.

**Figure 4 antioxidants-12-01874-f004:**
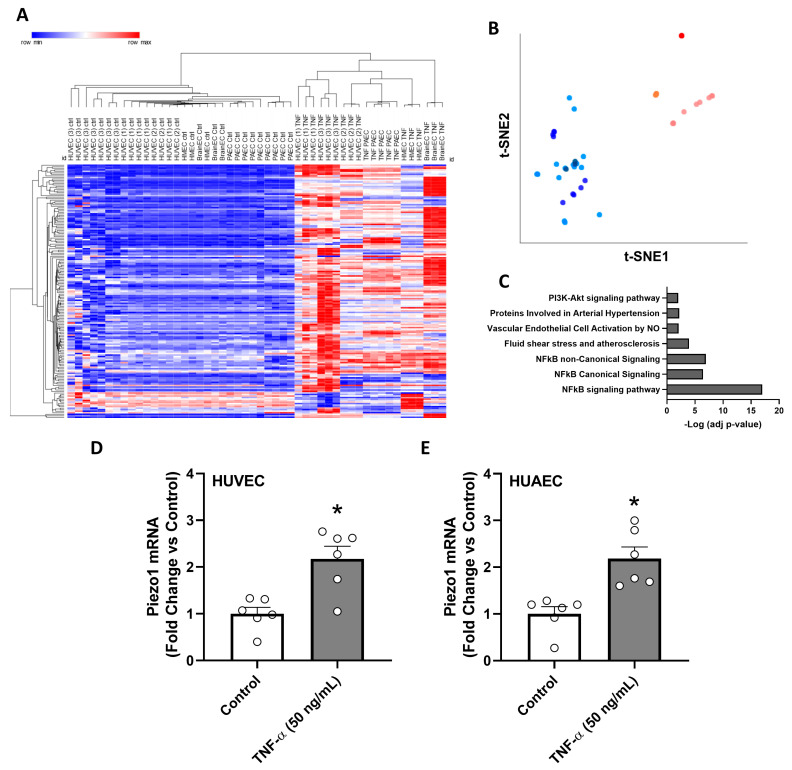
Effects of TNF-α stimulation on MRG and PIEZO1 transcript levels in endothelial cells from different vascular origins. (**A**) Hierarchical clustering of diverse endothelial cells exposed to different TNF-α treatments; (**B**) t-SNE projection showing clustering between diverse endothelial cell types and conditions. Endothelial cells exposed to TNF-α are represented in red, orange, and pink colors. Endothelial cells under basal conditions (control) are represented in blue and light blue colors. (**C**) Graph bars show seven enriched pathways in response to TNF-alpha treatment in diverse endothelial cells. (**D**,**E**) Fold change of PIEZO1 transcript levels in response to TNF-α treatment (50 ng/mL) for 24 h in HUVEC and HUAEC, respectively. Values are expressed as mean ± S.E.M., * *p* < 0.05 to control, Mann–Whitney U test.

**Figure 5 antioxidants-12-01874-f005:**
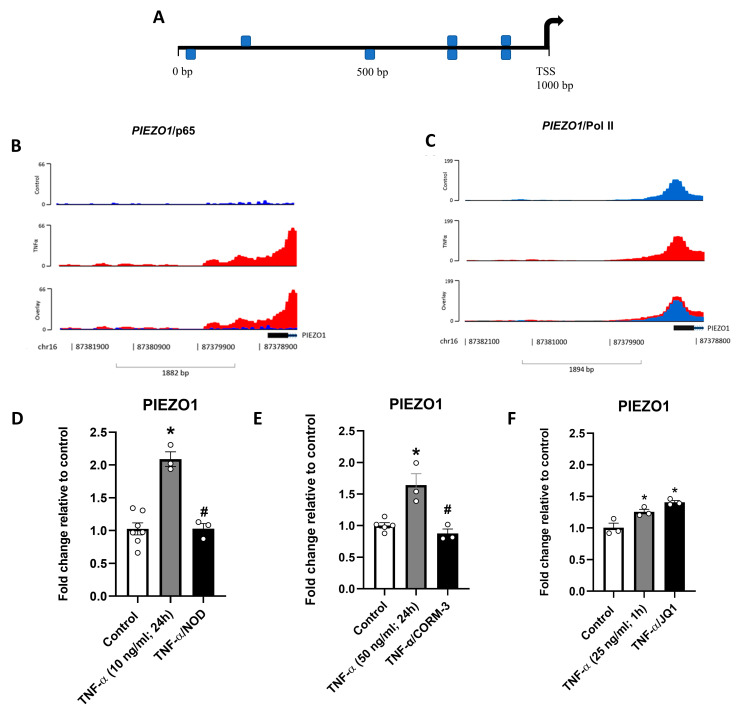
Interaction of PIEZO1/RELA and the effects of NF-kB inhibitors on TNF-α-induced PIEZO1 expression. (**A**) Predicted binding sites for NF-κB in the PIEZO1 promoter region zone (TSS: transcription start site; 1000 bp). (**B**) ChIP-seq diagram representing p65 interaction with PIEZO1 promoter region in control and treated cells. The overlay shows graphical differences between conditions. (**C**) ChIP-seq diagram representing Pol II interaction with PIEZO1 promoter in control and treated cells. The overlay shows no qualitative differences between conditions. (**D**) Fold change in PIEZO1 transcript levels in response to TNF-α treatment (10 ng/mL) for 24 h alone or with NOD, a p65 inhibitor. (**E**) Fold change in PIEZO1 transcript levels in response to TNF-α treatment (50 ng/mL) for 24 h alone or with CORM-3, a CO-donor, and p65 inhibitor. (**F**) Fold change in PIEZO1 transcript levels in response to TNF-α treatment (25 ng/mL) for 1 h alone or with JQ1, a BRD4 inhibitor. Values are expressed as mean ± S.E.M., * *p* < 0.05 to control, # *p* < 0.05 to TNF-α treated, one-way ANOVA, FDR post hoc test.

**Figure 6 antioxidants-12-01874-f006:**
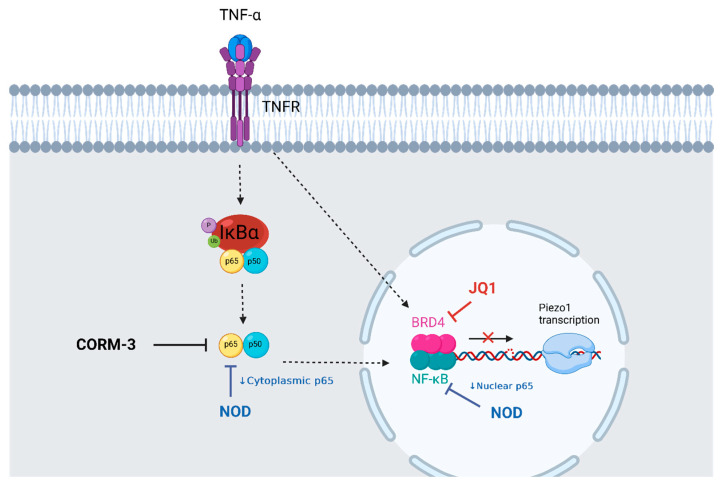
Summary of proposed PIEZO1 regulation by pro-inflammatory conditions in in vitro endothelial cells. TNF-α receptor activation by TNF-α led to NF-κB complex p65-p50 subunit activation and translocation to the nucleus after IkB-alpha ubiquitination. Cytoplasmic levels and activity of p65 are reduced by NOD and CORM-3, respectively. NOD also reduces nuclear p65 levels. BRD4, a transcriptional NF-κB enhancer, is inhibited by JQ1, but no effect in the regulation of PIEZO1 transcription was found. This diagram is simplified and does not consider every TNF-α signaling pathway (created in Biorender.com).

**Table 1 antioxidants-12-01874-t001:** Piezo1 and mechanosensing-related transcript levels in response to TNF-α treatment in diverse endothelial cell types.

Gene	Endothelial Cell Type	Fold-Change (Mean ± S.E.M)	FDR *p*-adj
PIEZO1	Brain	1.63 ± 0.05	**0.35**
	Pulmonary artery	1.75 ± 0.09	<0.01
	Skin microvascular	1.83 ± 0.04	<0.01
KLF4	Brain	0.99 ± 0.01	**0.99**
	Pulmonary artery	2.95 ± 0.13	<0.01
	Skin microvascular	1.15 ± 0.04	**0.09**
YAP	Brain	1.10 ± 0.04	**0.90**
	Pulmonary artery	0.66 ± 0.06	<0.01
	Skin microvascular	1.34 ± 0.11	**0.11**
NFKB1	Brain	17.71 ± 1.19	<0.01
	Pulmonary artery	3.85 ± 0.13	<0.01
	Skin microvascular	3.12 ± 0.01	<0.01
NFKB2	Brain	71.59 ± 2.53	<0.01
	Pulmonary artery	5.17 ± 0.34	<0.01
	Skin microvascular	3.72 ± 0.29	<0.01

Non-significant changes in bold.

## Data Availability

All the new material generated by this study is provided as Appendix A. All the dataset(s) supporting the conclusions of this article are available in the GEO Omnibus repository (https://www.ncbi.nlm.nih.gov/geo/, accessed on 30 October 2021), and Appendix A contains the download URL for each dataset.

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
