# Peer review of "Transcriptional Profiling of Human Endothelial Cells Unveils PIEZO1 and Mechanosensitive Gene Regulation by Prooxidant and Inflammatory Inputs"

_antioxidants, 2023, doi:10.3390/antiox12101874_

Round 1

Reviewer 1 Report

11)      This is a well-done study characterizing transcript regulation of Piezo1 gene in human endothelial cells. However, none of the analyses shows expression of Piezo1 protein. Most of the previous studies analyzing Piezo1 have been done in mouse models and many of them are also focused only on mRNA. At present, there is no evidence known to this reviewer that demonstrates expression of Piezo1 protein in blood vessels in normal or abnormal human tissues. The Human Protein Atlas website (https://www.proteinatlas.org/ENSG00000103335-PIEZO1/tissue) shows that although all human tissues express mRNA of Piezo1, only neuropils in the brain express Peizo1 protein while all other cells including endothelial cells are 100% negative. All cells in all human cancers including blood vascular endothelial cells are also 100% negative for expression of Piezo1 protein, although all of them express mRNA. This is a good time to ask the question: why should we be concerned about Piezo1 gene that cannot have any functional impact on human tissue physiology because it is not translated into protein in any human tissue except in neuropils of the brain. If only mice express endothelial Piezo1 protein, then it is a mouse-specific phenomenon. If only cultured human endothelial cells express this protein (which is not shown here), then it is a tissue culture restricted phenomenon. If Piezo1 protein is expressed in normal or pathological human tissues in blood vessels or any other structures or cell type, this has to be clearly shown by immunohistochemistry using double staining to identify a specific cell type positive for Piezo1. Without this evidence, description of the present data should be modified to indicate that authors’ interpretation refers exclusively to Peizo1 transcript levels in cultured endothelial cells.

22) Please do immunostaining and show expression of Piezo1 protein in blood vessels of any human tissue of choice.

Author Response

We thank reviewer for the valuable comments, which were addressed in the MS and the attached document

Reviewer 2 Report

This is an interesting paper which outlines the a study  that examined the transcriptional profiles of human endothelial cells tin order to determine  altered shear stress patterns and subsequent prooxidant and inflammatory conditions on Piezo1 and mechanosensitive-related genes (MRG).

1. I would have liked a summary diagram to outline the findings of the study.

2. Although PIEZO1 expression has been related to the pro-oxidant regulation what was the evidence that the cell was experiencing oxidative stress?

3. This study only really examines gene expression with no actual assessment of protein expression. The authors should include some information about the limitations of this study in the discussion section. 

4. What are the main pro-inflammatory conditions that of PIEZO1 and MRG pertinent to and would oxidative stress/reactive oxygen species per se be sufficient to induce the expression of the mechanosensitive genes? 

Author Response

We thank reviewer #2 for the valuable comments. 

Round 2

Reviewer 1 Report

The authors addressed my concerns.